# A Separable Self-attention Inspired by the State Space Model for Computer Vision

## Abstract

Separable self-attention is an early attention mechanism with linear complexity. When parameters and FLOPs are comparable, lightweight networks built upon separable self-attention and its variants underperform the recent Vision Mamba (ViM). By analyzing the strengths and weaknesses of separable self-attention, we distill four design principles and, inspired by the State Space Model (SSM) serving as the core of ViM, propose a novel separable self-attention termed Vision Mamba Inspired Separable self-Attention (VMI-SA). Notably, VMI-SA does not incorporate any SSM blocks, and its attention computation process differs from all existing attention mechanisms to the best of our knowledge. We introduce proof-of-concept networks, VMINet and VMIFormer, enabling fair comparisons with ViMs through deliberate control of parameters, FLOPs, and encoder numbers. Compared to state-of-the-art Transformers, CNNs, and ViMs, VMINet and VMIFormer achieve competitive results in image classification and high-resolution dense prediction tasks.

## 1 Introduction

Modern State Space Models (SSMs) excel at capturing long-range dependencies and reap the benefits of parallel training. The Vision Mamba (ViM) methods Zhu et al. (2024a); Liu et al. (2024); Huang et al. (2024); Pei et al. (2025), which are inspired by recently proposed SSMs Gu & Dao (2023); Mehta et al. (2023), utilize the Selective Space State Model (S6) to compress previously scanned information into hidden states, effectively reducing quadratic complexity to linear. Many studies integrate the original SSM framework from Mamba into their foundational models to balance performance and computational efficiency. However, Mamba is not the first model to achieve global modeling with linear complexity. Linear attention Katharopoulos et al. (2020) replaces the non-linear softmax function with linear normalization and adds a kernel function to both query and key, allowing for the reordering of computation based on the associative property of matrix multiplication, thereby reducing the computational complexity to linear. Separable self-attention Mehta & Rastegari (2023) is also an early work that replaces the computationally expensive operations (e.g., batch-wise matrix multiplication) in Multi-headed Self-Attention (MHA) with element-wise operations (e.g., summation and multiplication). However, because of the limited expressive capabilities of separable self-attention and its variants, they are typically suitable for lightweight vision Transformers that have been carefully designed.

In vision tasks, a prevalent belief posits an inherent conflict between the non-causal nature of 2D spatial patterns and the unidirectional causality of SSMs. Flattening spatial data into 1D tokens destroys the local 2D dependencies in the image, thereby impairing the model's capacity to accurately interpret spatial relationships. Vim Zhu et al. (2024a) addresses this issue by scanning in bidirectional horizontal directions. Subsequent works, such as LocalMamba Huang et al. (2024) and EfficientVMamba Pei et al. (2025), have designed a series of novel scanning strategies. These efforts aim to expand the receptive field of the SSM from the previous token to others, which may result in a multiple-fold increase in the computational cost of the scanning process. However, some work has questioned the necessity of complex scanning patterns. Liu et al. (2024) found that even with the simplest unidirectional scanning strategy, the performance of VMamba is not significantly impacted. Zhu et al. (2024b) conducted a comprehensive experimental investigation on the impact of mainstream scanning directions and their combinations on semantic segmentation of remotely sensed images. Through extensive experiments on the LoveDA, ISPRS Potsdam, and ISPRS Vai-

hingen datasets, they demonstrate that no single scanning strategy outperforms others, regardless of their complexity or the number of scanning directions involved. Given that ViMs generally outperform other early linear models, we attribute this to the introduction of causality: specifically, the SSM's causal framework preserves diverse local correlations during global information compression.

This paper first establishes design principles for Vision Mamba Inspired Separable self-Attention (VMI-SA) by analyzing the advantages and limitations of separable self-attention mechanisms versus Softmax self-attention. Subsequently, drawing inspiration from ViM, we design an autoregressive model for encoding visual information, termed the recurrent formulation of VMI-SA. This formulation employs masking to encode multi-scale historical context into fixed-length context vectors. Compared to the original separable self-attention, our approach more effectively models dependencies across tokens. Finally, by eliminating the receptive field limitation, recurrent formulation of VMI-SA can be implemented entirely via parallelizable matrix operations, thereby maintaining the computational efficiency inherent to separable self-attention.

We propose two proof-of-concept networks, VMINet and VMIFormer, based on VMI-SA. Through deliberate control over parameters, FLOPs, and the number of encoders, we perform fair comparisons with ViM models employing simple architectures (e.g., Vim Zhu et al. (2024a) and PlainMamba Yang et al. (2024)) and with those utilizing Transformer architectures or complex hybrid architectures (e.g., VMamba Liu et al. (2024) and MambaVision Hatamizadeh & Kautz (2025)).

## 2 PRELIMINARIES

This section briefly reviews the basic forms of Self-Attention, Separable Self-Attention, and Structured State Space Model.

### 2.1 SOFTMAX SELF-ATTENTION

In a broad sense, attention refers to a computational process that dynamically allocates importance weights to different information sources according to the needs of the current task, thereby forming a more meaningful representation through information aggregation. The most widely used and significant variant of attention is the softmax self-attention, which can be defined as:

$$Y = softmax(QK^T) \cdot V \tag{1}$$

where $Q, K, V \in \mathbb{R}^{(L,D)}$ respectively represent $L$ tokens with $D$ dimensions, each generated by a linear transformation from the input $X \in \mathbb{R}^{(L,C)}$. The attention scores between each pair of tokens in $Q$ and $K$ are computed using the dot product operation. Subsequently, interactions are normalized using softmax. Finally, the weighted interactions are multiplied by $V$ using the dot product operation to produce the final weighted output. The pairwise comparison mechanism, realized by computing $QK^T$, results in a quadratic growth in the attention's training cost. The entire computation process is illustrated in Figure 1 (d).

### 2.2 SEPARABLE SELF-ATTENTION

The structure of separable self-attention is inspired by Softmax Self-Attention Mehta & Rastegari (2023). Similar to softmax self-attention, the input $X \in \mathbb{R}^{(L,C)}$ is processed using three branches: $Q \in \mathbb{R}^{(L,1)}, K \in \mathbb{R}^{(L,D)}$ and $V \in \mathbb{R}^{(L,D)}$. Notably, $Q$ maps each token in $X$ to a scalar, distinguishing it from the other branches. First, context scores are generated through $Softmax(Q)$. Then, based on broadcasting mechanism, the context scores are then element-wise multiplied with $K$ and the resulting vector is summed over the token dimension to obtain the context vector. Finally, the context vector is multiplied by $V$ using broadcasted element-wise multiplication to spread the contextual information and produce the final output. It can be summarized as:

$$Y = \sum_{i=1}^{L} \left( softmax(Q) \odot K \right)_i \odot V \tag{2}$$

Here, $\odot$ denotes element-wise multiplication. The process follows the broadcasting mechanism throughout, as illustrated in Figure 1 (c).

## 2.3 STRUCTURED STATE SPACE MODEL

Structured State Space Sequence Model (S4) is a recent sequence model for deep learning, which is widely related to RNNs, CNNs, and classical SSMs. Their inspiration stems from a specific continuous system that, through an implicit latent state $h \in \mathbb{R}^{(D,L)}$, maps a one-dimensional sequence $x \in \mathbb{R}^L$ to another one-dimensional sequence $y \in \mathbb{R}^L$ Dao & Gu (2024). The mapping process could be denoted as:

$$
\begin{aligned}
h_i &= A h_{i-1} + B x_i \\
y_i &= C^{\mathrm{T}} h_i
\end{aligned}
\tag{3}
$$

where $i \in [1, L]$, $A \in \mathbb{R}^{(D,D)}$, $B \in \mathbb{R}^{(D,1)}$ and $C \in \mathbb{R}^{(D,1)}$. The Selective State Space Model (S6) adopted by Mamba Gu & Dao (2023) is developed based on it. In this paper, we use the term state space model (SSM) to refer to various variants of SSMs, including S4 and S6.

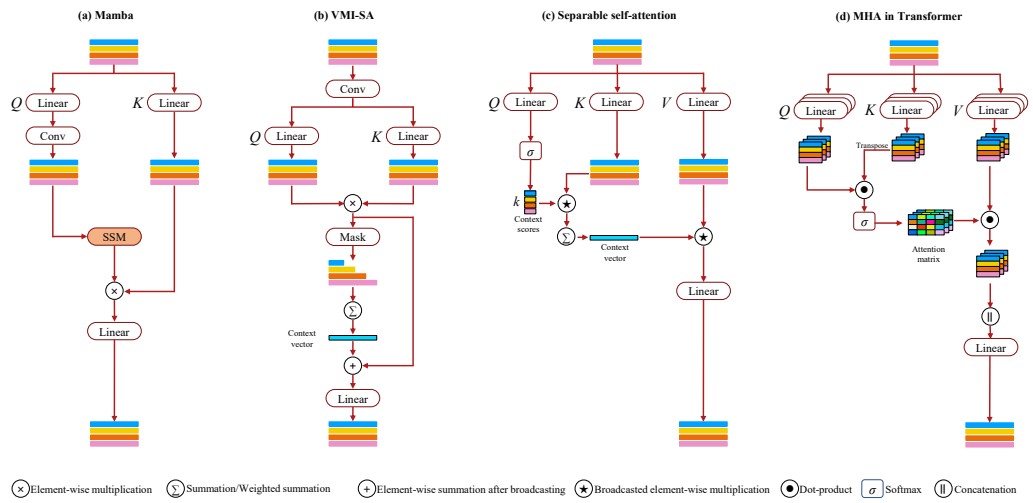

Figure 1: Comparison with different modules. To facilitate a clear comparison, we uniformly adapt one-dimensional sequences as input, although this is not necessary for VMI-SA.

## 3 METHODOLOGY

In this section, we first analyze the impact of the key differences in design between separable self-attention and softmax self-attention. Then, while retaining the advantages of the self-attention design, we optimize the separable self-attention according to the design method of SSM. Our goal is to clearly demonstrate the design process of Vision Mamba Inspired Separable Self-Attention (VMI-SA), to show the innovations and how performance can be enhanced by integrating the strengths of both Mamba and separable self-attention. Finally, we introduce the overall architecture of the proof-of-concept networks, VMINet and VMIFormer.

### 3.1 ELEMENT-WISE MULTIPLICATION INSTEAD OF MATRIX MULTIPLICATION

In both traditional machine learning and deep learning, handling features in high-dimensional space is crucial. We employ a straightforward derivation to establish that both element-wise multiplication and matrix multiplication can map the features from their original dimensions to a higher-dimensional space, which is crucial for feature representation.

We adopt the definition method from Section 2, let $X \in \mathbb{R}^{(L,C)}, W^1 \in \mathbb{R}^{(C,D)}, W^2 \in \mathbb{R}^{(C,D)}, Q = XW^1, K = XW^2, E = Q \odot K$. For any element $E_{m,n}$ in $E$ (where $m \in [1, L]$, and $n \in [1, D]$):

$$
\begin{aligned}
E_{m,n} &= Q_{m,n} \times K_{m,n} \\
&= \big( \sum_{i=1}^{C} X_{m,i} W_{i,n}^1 \big) \times \big( \sum_{j=1}^{C} X_{m,j} W_{j,n}^2 \big) \\
&= \sum_{i=1}^{C} \sum_{j=1}^{C} W_{i,n}^1 W_{j,n}^2 X_{m,i} X_{m,j} \\
&= \underbrace{a_{(1,1)} X_{m,1} X_{m,1} + \cdots + a_{(C,C)} X_{m,C} X_{m,C}}_{C(C+1)/2 \text{ items}}
\end{aligned}
\tag{4}
$$

where $a$ is a coefficient for each item:

$$
a_{(i,j)} = \begin{cases} W_{i,n}^1 W_{j,n}^2 & \text{if } i = j, \\ W_{i,n}^1 W_{j,n}^2 + W_{j,n}^1 W_{i,n}^2 & \text{if } i \neq j \end{cases}
\tag{5}
$$

Each term in Eq. (4) exhibits a nonlinear relationship with the input. It can be approximated as that the element-wise multiplication operation projects the feature vector in the $C$-dimensional space into a higher-dimensional space of $C^2$ dimensions through a nonlinear transformation and processes it.

Now let's discuss the case of matrix multiplication. Let $E' = Q \cdot K^{\mathrm{T}}$, where any element $E'_{m,n}$:

$$
\begin{aligned}
E'_{m,n} &= \sum_{t=1}^{D} Q_{m,t} \times K_{t,n}^{\mathrm{T}} \\
&= \sum_{t=1}^{D} \left[ \big( \sum_{i=1}^{C} X_{m,i} W_{i,t}^1 \big) \times \big( \sum_{j=1}^{C} W_{j,t}^2 X_{n,j} \big) \right] \\
&= \sum_{t=1}^{D} \sum_{i=1}^{C} \sum_{j=1}^{C} W_{i,t}^1 W_{j,t}^2 X_{m,i} X_{n,j}
\end{aligned}
\tag{6}
$$

By comparing Eq. (4) and Eq. (6), we observe that both the element-wise product (with linear computational cost) and the matrix multiplication (with quadratic computational cost) can be viewed as operations that non-linearly map feature vectors from a $C$-dimensional space into a space of approximately $C^2$ dimensions. From this perspective, the element-wise product is more efficient.

### 3.2 CONTEXT VECTOR INSTEAD OF ATTENTION MATRIX

The context vector in Eq. (2) is analogous to the attention matrix $softmax(QK^{\mathrm{T}})$ in a sense that it also encodes the information from all tokens in the input $X$ Mehta & Rastegari (2023), but is cheap to compute. Comparing Eq. (4) and Eq. (6), it can be observed that $E_{m,n}$ is merely the encoding of the $m$-th token, while $E'_{m,n}$ is the encoding of both the $m$-th and $n$-th tokens. The softmax and summation operations provide a global receptive field for separable self-attention, but the performance difference between separable self-attention and softmax self-attention indicates that establishing correlations between tokens is essential. We speculate that this is also the reason why networks adopting separable self-attention or its variants, such as MobileViT Mehta & Rastegari (2023) and SwiftFormer Shaker et al. (2023), need to alternately stack the attention modules with local feature encoding modules and feedforward neural network modules. In fact, this perspective is also supported by evidence in ViMs. The SSM restricts the receptive field to the previous token, yet it is still applicable for visual tasks. In addition, it is easy to observe from Eq. (2) and Eq. (4) that, due to the parameter sharing across different tokens, the simple summation operation results in identical weights for each token in the global context information, thereby making the computation process of Eq. (2) lack "attention". Therefore, in Eq. (2), the context vector is element-wise multiplied with $V$, which, aside from mapping features to a higher dimension, does not have much clear significance. Moreover, ShuffleNet Ma et al. (2018) points out that while "multi-path" structured

network blocks can enhance accuracy, they introduce additional overhead such as kernel launches and synchronization, thereby affecting efficiency. Thus, we argue that employing the same "three-branch" structure in separable self-attention as in softmax self-attention is unnecessary.

Additionally, we can analyze the performance differences between softmax self-attention and separable self-attention from the perspective of the rank of the attention matrix. The higher the rank of the attention matrix, the more attention information it contains, and the richer the feature diversity. FLatten Transformer Han et al. (2023) incorporated an additional attention computation branch (Depthwise Convolution) to the linear attention mechanism and visualized changes in the rank of attention matrices. Experiments demonstrate that this enhancement enables the attention matrix to achieve full rank, resulting in significant performance improvements for the model. This indicates a positive correlation between the attention matrix rank and model performance. The attention matrix $softmax(QK^{\mathrm{T}})$ in Eq. (1) is usually full rank Han et al. (2023), that is $\mathrm{rank}(softmax(QK^{\mathrm{T}})) = L$. The attention information in the context vector comes from $softmax(Q) \odot K$ in Eq. (2), and its rank:

$$\mathrm{rank}(softmax(Q) \odot K) \leq \mathrm{rank}(K) \leq \min\{L, D\}. \tag{7}$$

Therefore, the attention information in $softmax(Q) \odot K$ is not only less abundant but also severely homogenized.

### 3.3 Vision Mamba Inspired Separable Self-Attention

Summarizing the analysis, the previous discussion provides the following four insights for the design of new separable self-attention:

- Continue to use element-wise multiplication for context encoding while reducing the computational branches.
- Introduce correlation between tokens.
- Enhancing the rank of attention matrices or equivalent counterparts.
- Utilize learnable weights to adjust the intensity of each token's contribution to the context information.

#### 3.3.1 Excellent Design in Mamba

Our analysis results show several similarities with the design philosophies of Mamba. As illustrated in Figure 1 (a), for a single Mamba block, the input is processed through two computational branches and then fused via element-wise multiplication, where one branch uses convolution to establish local correlations.

In addition, Mamba preserves and compresses global information through the SSM module, which is analogous to the $softmax(QK^{\mathrm{T}})$ in softmax self-attention mechanism but with linear complexity. As an RNN-based model, Mamba is sensitive to the order of the input sequence, and its scanning process provides the model with positional information. Therefore, unlike transformers, Mamba does not require additional positional encoding.

#### 3.3.2 Macro Design

Our objective is to implement the aforementioned four design philosophies using the simplest and most direct approach, thereby improving the original separable self-attention mechanism without introducing superfluous functional blocks. First, adhering to the design philosophy of separable self-attention, we still utilize context vectors to represent global information. Second, since the contextual information is generated through element-wise multiplication, there is no need to flatten 2D image data into a one-dimensional sequence. Compared to some common Transformers and ViMs, processing features in 2D space can maintain the spatial correlation of features, avoiding the additional inductive bias introduced by Patch Embedding. Additionally, it can reduce the reshaping operations, which is beneficial for improving the inference speed. As previously mentioned, element-wise multiplication can encode the features for individual tokens in pairs, but it cannot establish correlations between tokens. Therefore, the simplest and most effective improvement is

to use a depthwise convolution (DW-Conv) layer to establish local spatial correlations before the element-wise multiplication.

Next, we consider how to enhance the rank of the attention matrix (or equivalent counterparts). Clearly, for any matrix $A \in \mathbb{R}^{(L,D)}$ with all elements being non-zero, assuming $L > D$, setting the elements of the upper triangular (or lower triangular) part of $A$ to zero can maximize the rank of the matrix, that is:

$$
M = \begin{bmatrix} 1 & & & \\ 1 & 1 & & \\ \vdots & \vdots & \ddots & \\ 1 & 1 & \cdots & 1 \\ \vdots & \vdots & \vdots & \vdots \\ 1 & 1 & \cdots & 1 \end{bmatrix},
\tag{8}
$$

$$
\mathrm{rank}(M \odot A) = \min\{L, D\} = D,
$$

where $M \in \mathbb{R}^{(L,D)}$. If the matrix $A$ equals the $softmax(Q) \odot K$ from Eq. (2) and $M$ is regarded as a causal mask matrix, an interesting conclusion can be drawn: the introduction of causality into the separable self-attention can theoretically increase the diversity of contextual information, thereby enhancing performance. Therefore, we believe that it is feasible to improve the separable self-attention by referring to Eq. (3).

### 3.3.3 RECURRENT FORMULATION

Han et al. (2024) pointed out that converting linear attention to causal linear attention and introducing a forget gate can significantly improve model performance on ImageNet-1K. It can be observed that in the shallow layers of the network, each token mainly focuses on itself and the two preceding tokens; as the network depth increases, the attention range of each token gradually enlarges. The work of Han et al. indicates that for attention mechanisms with linear computational complexity, the combination of local and global information contributes to forming more effective attention, although their contributions vary at different stages.

Similar to Eq. (3), we restrict the receptive field to the previous token and preserve past information through a hidden state. Given the non-causal nature of image data and the shorter sequence lengths being processed, we argue that VMI-SA should preserve all historical information rather than decaying it through matrix $A$ as in Mamba. The recurrent formulation of the VMI-SA is as follows:

$$
\begin{aligned}
h_i &= h_{i-1} + \alpha_i(Q_i \odot K_i) \\
y_i &= M_i \odot h_i + \beta_i(Q_i \odot K_i)
\end{aligned}
\tag{9}
$$

where $X \in \mathbb{R}^{(H,W,C)}, W^1 \in \mathbb{R}^{(C,D)}, W^2 \in \mathbb{R}^{(C,D)}, Q = \mathrm{DW} - \mathrm{Conv}(X)W^1, K = \mathrm{DW} - \mathrm{Conv}(X)W^2, L = H * W, i \in [1, L], M \in \mathbb{R}^{(L,D)}$ is a lower triangular matrix with all non-zero elements equal to 1, $\alpha_i$ and $\beta_i$ are a series of trainable parameters that control the importance of each token in contextual information, as well as the proportion of local information to contextual information in attention. Like Mamba, we also do not use softmax. $M_i \odot h_i$ in Eq. (9) can be regarded as the context vector $\mathbf{c_v}$ of VMI-SA. In the original separable self-attention, each element of the context vector encodes information about all tokens, whereas in VMI-SA, the $i$-th element encodes information about the $i$-th token and all subsequent tokens. Consequently, VMI-SA's context vector more effectively characterizes dependencies among tokens at varying scales.

### 3.3.4 MATRIX FORMULATION

Similar to RNN-based models, the recurrent formulation of VMI-SA is not computationally efficient. The main reason that prevents VMI-SA from being implemented via parallelizable matrix operations is that each token can only utilize information from tokens that precede it in the sequence. Therefore, we remove the restriction on the receptive field and allow all tokens to receive the same

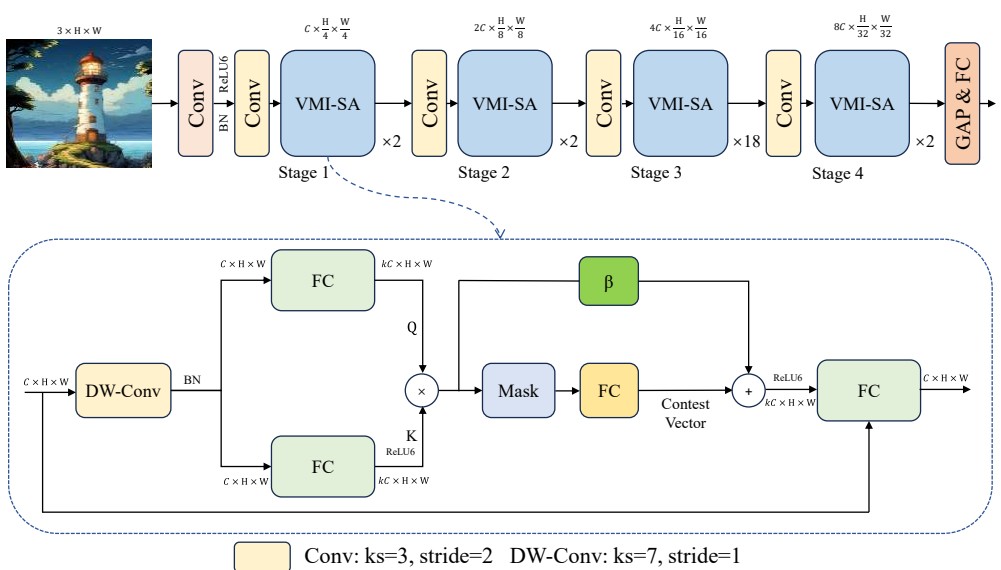

Figure 2: VMINet architecture overview.

global information. Eq. (9) is transformed into:

$$
\begin{cases}
Y = \mathrm{Expand}_L\left(\sum_{i=1}^{L} \alpha_i \cdot M_i \odot Q_i \odot K_i\right) + \beta \cdot Q \odot K \\
\mathbf{c_v} = \sum_{i=1}^{L} \alpha_i \cdot M_i \odot Q_i \odot K_i
\end{cases}
\tag{10}
$$

where $\mathrm{Expand}_L(\cdot)$ denotes the operation of expanding a vector of shape $(1, D)$ into a matrix of shape $(L, D)$, $\mathbf{c_v}$ is the context vector of VMI-SA. The primary network structure of VMI-SA is shown in Figure 1 (b).

### 3.4 VMINet AND VMIFormer

As shown in Figure 2, VMINet adopts a common 4-stage hierarchical architecture, utilizing convolutional layers for downsampling, and employing VMI-SA blocks for feature extraction. To ensure a fair comparison with the Vim Zhu et al. (2024a), which uses a pure Mamba encoder, we set the number of VMI-SA blocks to be the same as the number of Mamba blocks with a comparable parameter count. More details can be found in Table 1. Additionally, while maintaining consistent feature map sizes, we constructed VMIFormer-T by replacing the SS2D block of VMamba-T Liu et al. (2024) with the VMI-SA block. Detailed architectural designs of VMIFormer-T are provided in the Appendix.

Table 1: Configurations of VMINet.

| Variant | $C$ | $k$ | Params |
|---|---|---|---|
| VMINet-Ti | 24 | 2 | 2.0M |
| VMINet-XS | 48 | 2 | 7.4M |
| VMINet-S | 48 | 4 | 13.3M |
| VMINet-B | 96 | 2 | 28.4M |

## 4 EXPERIMENTS

This section presents our experimental results, starting with the ImageNet classification task and then transferring the trained model to various downstream tasks, including object detection, instance segmentation. Additional experimental settings and supplementary experiments are provided in the Appendix.

### 4.1 IMAGE CLASSIFICATION ON IMAGENET-1K

**Settings.** We train the models on ImageNet-1K and evaluate the performance on ImageNet-1K validation set. For fair comparisons, our training settings mainly follow Vim Zhu et al. (2024a). Unlike Vim, our experiments are performed on 3 A6000 GPUs. Therefore, we adjusted the total batch size and the initial learning rate to 384 and $5 \times 10^{-4}$ respectively.

Table 2: Comparison of different models on ImageNet-1K. †: In contrast with most of the work presented in the table, MobileViTv2 utilizes a larger resolution of $256 \times 256$, while SwiftFormer employs knowledge distillation.

| Method | Params (M) | FLOPs (G) | Top-1 (%) |
| --- | --- | --- | --- |
| MobileViTv2-0.5† (TMLR 2023) | 1 | 0.5 | 70.2 |
| PVTv2-B0 (CVM 2022) | 3 | 0.6 | 70.5 |
| VMINet-Ti **(ours)** | **2** | **0.3** | **70.7** |
| EfficientViT-M2 (CVPR 2023) | 4 | 0.2 | 70.8 |
| LVT (CVPR 2022) | 6 | 0.9 | 74.8 |
| Vim-Ti (ICML 2024a) | 7 | 1.5 | 76.1 |
| FasterNet (CVPR 2023) | 8 | 0.9 | 76.2 |
| LocalVim-T (ECCV 2024) | 8 | 1.5 | 76.5 |
| MobileOne-S2 (CVPR 2023) | 8 | 1.3 | 77.4 |
| PlainMamba-L1 (BMVC 2024) | 7 | 3.0 | 77.9 |
| MobileMamba-S6 (CVPR 2025) | 15 | 0.6 | 78.0 |
| MobileViTv2-1.0† (TMLR 2023) | 5 | 1.8 | 78.1 |
| StarNet-S4 (CVPR 2024) | 8 | 1.1 | 78.4 |
| SwiftFormer-S† (ICCV 2023) | 6 | 1.0 | 78.5 |
| DefMamba-T (CVPR 2025) | 8 | 1.2 | 78.6 |
| VMINet-XS **(ours)** | **7** | **1.4** | **78.6** |
| EfficientVMamba-S (AAAI 2025) | 11 | 1.3 | 78.7 |
| VCMamba-S (ICCV 2025) | 11 | 2.2 | 78.7 |
| DeiT-S (ICML 2021) | 22 | 4.6 | 79.8 |
| RegNetY-4G (CVPR 2020) | 21 | 4.0 | 80.0 |
| MobileViTv2-1.5† (TMLR 2023) | 11 | 4.0 | 80.4 |
| Vim-S (ICML 2024a) | 26 | 5.1 | 80.5 |
| VMINet-S **(ours)** | **13** | **2.3** | **80.5** |
| SwiftFormer-L1† (ICCV 2023) | 12 | 1.6 | 80.9 |
| LocalVim-S (ECCV 2024) | 28 | 4.8 | 81.0 |
| Swin-T (CVPR 2021) | 29 | 4.5 | 81.3 |
| VCMamba-M (ICCV 2025) | 21 | 4.6 | 81.5 |
| PlainMamba-L2 (BMVC 2024) | 25 | 8.1 | 81.6 |
| EfficientVMamba-B (AAAI 2025) | 33 | 4.0 | 81.8 |
| ConvNeXt-T (CVPR 2022) | 29 | 4.5 | 82.1 |
| MambaVision-T (CVPR 2025) | 32 | 4.4 | 82.3 |
| VMINet-B **(ours)** | **28** | **4.8** | **82.4** |
| VMamba-T (NeurIPS 2024) | 30 | 4.9 | 82.6 |
| VCMamba-B (ICCV 2025) | 32 | 8.0 | 82.6 |
| VMIFormer-T **(ours)** | **24** | **4.2** | **83.2** |

**Results.** We selected advanced CNNs, ViTs, and ViMs with comparable parameters and computational costs to compare with our method, and the results are shown in Table 2. Experimental results demonstrate that lightweight models constructed with VMI-SA can compete with state-of-the-art counterparts across various parameter scales and FLOPs. PlainMambaYang et al. (2024) has two variants, L1 and L2, which adopt the same configuration of 24 blocks as Vim and VMINet, and employ depthwise convolutions to establish 2D local correlations before selective scanning. Compared with PlainMamba, our VMINet exhibits significant advantages in terms of performance,

efficiency, and model complexity. VMIFormer-T significantly outperforms VMamba-T Liu et al. (2024), demonstrating VMI-SA's adaptability to advanced network architectures. MambaVision is a four-stage backbone network that adopts a hybrid architecture, designed for high performance. The first two stages utilize convolutional blocks to achieve rapid feature extraction, while the last two stages employ both MambaVision blocks and Transformer blocks to capture long-range spatial dependencies Hatamizadeh & Kautz (2025). As models with linear complexity, VMINet-B and VMIFormer-T outperform MambaVision-T, which has non-linear complexity.

## 4.2 OBJECT DETECTION AND INSTANCE SEGMENTATION ON COCO

**Settings.** We use Mask-RCNN as the detector to evaluate the performance of the proposed VMINet for object detection and instance segmentation on the MSCOCO 2017 dataset. Following ViTDetLi et al. (2021), we only used the last feature map from the backbone and generated multi-scale feature maps through a set of convolutions or deconvolutions to adapt to the detector. The remaining settings were consistent with SwinLiu et al. (2021).

**Results.** In Table 3, we summarize the comparative results of our method against other backbone networks. Similar to the classification task outcomes, VMINet, with its simple architecture, achieves a favorable trade-off among performance, parameter count, and computational cost, yielding results comparable to state-of-the-art ViMs, CNNs, and ViTs. Compared to Vmamba, VMIFormer exhibits a further enhanced performance advantage in high-resolution dense prediction tasks. We attribute this primarily to SSM's strong focus on modeling and compressing global information, which discards certain image details, whereas VMI-SA better balances the relationship between local and global information, capturing richer local semantic information.

Table 3: Object detection and instance segmentation results on COCO.

| Backbone | Params | FLOPs | $AP^b$ | $AP^b_{50}$ | $AP^b_{75}$ | $AP^m$ | $AP^m_{50}$ | $AP^m_{75}$ |
|---|---|---|---|---|---|---|---|---|
| ResNet-18 (CVPR 2016) | 31M | 207G | 34.0 | 54.0 | 36.7 | 31.2 | 51.0 | 32.7 |
| Vim-Ti (ICML 2024a) | 27M | 189G | 36.6 | 59.4 | 39.2 | 34.9 | 56.7 | 37.3 |
| PVT-T (ICCV 2021) | 33M | 208G | 36.7 | 59.2 | 39.3 | 35.1 | 56.7 | 37.3 |
| ResNet-50 (CVPR 2016) | 44M | 260G | 38.0 | 58.8 | 41.4 | 34.7 | 55.7 | 37.2 |
| VMINet-XS (**ours**) | 27M | 189G | 38.9 | 61.9 | 42.4 | 36.4 | 58.7 | 38.8 |
| EfficientVMamba-S (AAAI 2025) | 31M | 197G | 39.3 | 61.8 | 42.6 | 36.7 | 58.9 | 39.2 |
| ResNet-101 (CVPR 2016) | 63M | 336G | 40.0 | 60.5 | 44.0 | 36.1 | 57.5 | 38.6 |
| Vim-S (ICML 2024a) | 44M | 272G | 40.9 | 63.9 | 45.1 | 37.9 | 60.8 | 40.7 |
| Swin-T (CVPR 2021) | 48M | 267G | 42.7 | 65.2 | 46.8 | 39.3 | 62.2 | 42.2 |
| VMINet-S (**ours**) | 32M | 201G | 43.2 | 65.3 | 47.3 | 39.3 | 62.2 | 42.3 |
| ConvNeXt-T (CVPR 2022) | 48M | 262G | 44.2 | 66.6 | 48.3 | 40.1 | 63.3 | 42.8 |
| VMamba-T (NeurIPS 2024) | 48M | 276G | 44.3 | 65.2 | 49.5 | 40.3 | 62.8 | 43.9 |
| VMINet-B (**ours**) | 47M | 276G | 44.5 | 66.7 | 48.6 | 40.5 | 63.7 | 43.7 |
| VMIFormer-T (**ours**) | 46M | 275G | 45.3 | 67.6 | 49.6 | 41.3 | 64.9 | 44.3 |

## 5 CONCLUSION

In this paper, we propose VMI-SA, a novel separable self-attention mechanism with linear complexity. Drawing inspiration from ViMs, we first establish local correlations using depthwise convolution. Subsequently, We restrict the receptive field to the previous token to integrate local information with global historical contexts using a recurrent model, thus establishing the recurrent formulation of VMI-SA. To leverage efficient matrix operations, we expand the receptive field to global scope, yielding the matrix formulation of VMI-SA. Building upon VMI-SA, we develop two network architectures: VMINet and VMIFormer. Under fair comparisons (by controlling model parameters, FLOPs, and encoder count), we evaluated our approach against ViMs across image classification, object detection, and instance segmentation tasks. Experimental results demonstrate that VMI-SA consistently outperforms SSMs on image-based vision tasks. We believe that our work offers a new perspective for the future design of attention mechanisms or visual backbone networks: by adjusting expressions and constraints within a unified theoretical framework, it becomes possible to integrate the advantages of different approaches while achieving a balance between performance and efficiency.

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

# A APPENDIX

In this section, we provide additional details regarding:

- Architecture Details of VMIFormer
- Datasets and Experiment Details
- Empirical studies on ImageNet-1K
- Additional Experimental Results

## A.1 ARCHITECTURE DETAILS OF VMIFORMER

An overview of the architecture of VMIFormer-T is illustrated in Figure 3(a). The input image is first partitioned into patches by a stem module, resulting in a 2D feature map with spatial dimension of $H/4 \times W/4$. Without incorporating additional positional embeddings, multiple network stages are employed to create hierarchical representations with resolutions of $H/8 \times W/8$, $H/16 \times W/16$, and $H/32 \times W/32$. Specifically, each stage comprises a downsampling layer (except for the first stage), followed by a stack of VMIFormer blocks. As shown in Figure 3(b) and (c), both the VMIFormer block and the VMamba's VSS block follow the design philosophy of the vanilla Transformer block, with the sole difference lying in their token mixers.

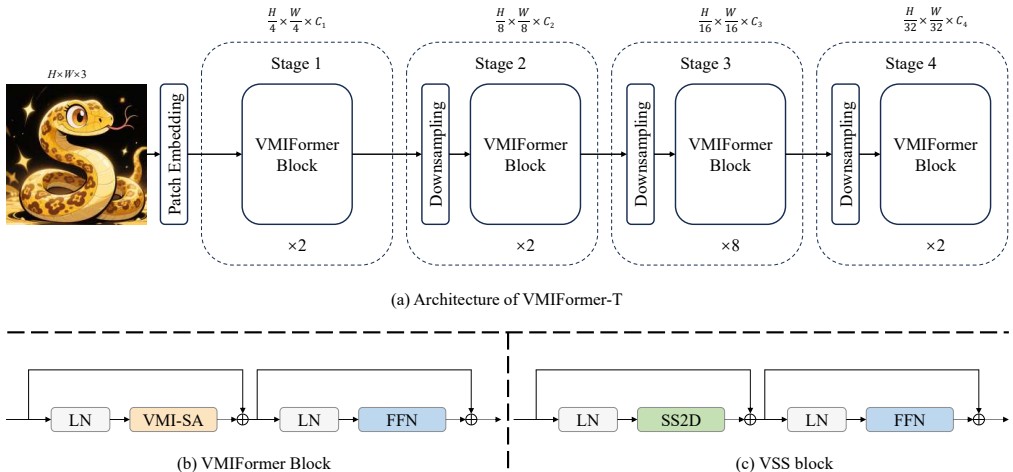

(a) Architecture of VMIFormer-T

(b) VMIFormer Block  (c) VSS block

Figure 3: Illustration of (a) Overall architecture of VMIFormer, (b) Structure of the proposed VMIFormer block, and (c) VSS block structure of VMamba Liu et al. (2024) for reference.

## A.2 DATASETS AND EXPERIMENT DETAILS

**ImageNet classification.** The ImageNet-1K dataset comprises 1.28 million training images and 50,000 validation images, encompassing 1,000 classes. For fair comparisons, our training settings mainly follow Vim Zhu et al. (2024a). Specifically, we apply random cropping, random horizontal flipping, label-smoothing regularization, mixup, and random erasing as data augmentations. When training on $224 \times 224$ input images, we employ AdamW with a momentum of 0.9 and a weight

decay of 0.025 to optimize models. During testing, we apply a center crop on the validation set to crop out $224 \times 224$ images. We train the VMINet and VMIFormer models for 300 epochs using a cosine schedule. Unlike Vim, our experiments are performed on 3 A6000 GPUs. Therefore, we adjusted the total batch size and the initial learning rate to 384 and $5 \times 10^{-4}$ respectively.

**COCO object detection.** MSCOCO 2017 dataset is a widely adopted benchmark for object detection and instance segmentation with 118K training and 5K validation images. We use Mask-RCNN as the detector to evaluate the performance of the proposed VMINet for object detection and instance segmentation on the MSCOCO 2017 dataset. Following ViTDetLi et al. (2021), we only used the last feature map from the backbone and generated multi-scale feature maps through a set of convolutions or deconvolutions to adapt to the detector. The remaining settings were consistent with SwinLiu et al. (2021). Specifically, we employ the AdamW optimizer and fine-tune the pre-trained classification models (on ImageNet-1K) for both 12 epochs ($1\times$ schedule). The learning rate is initialized at $1 \times 10^{-4}$ and is reduced by a factor of $10\times$ at the 9th and 11th epochs.

**ADE20K semantic segmentation.** ADE20K dataset contains 25K images, 20K for training, 2K for validation, and 3K for testing, with 150 semantic categories. Following Vim Zhu et al. (2024a), we train UperNet Xiao et al. (2018) with our VMINet on ADE20K dataset. In training, we employ AdamW with a weight decay of 0.01, and a total batch size of 16 to optimize models. The employed training schedule uses an initial learning rate of $6 \times 10^{-5}$, linear learning rate decay, a linear warmup of 1500 iterations, and a total training of 160K iterations.

### A.3 Empirical Studies on ImageNet-1K

**Recurrent formulation vs. matrix formulation.** Given that the computational complexity difference between the matrix formulation and the recurrent formulation of VMI-SA is negligible, we use latency to measure the actual runtime efficiency difference between them. For comparison, we also report the results of MobileViTv2 Mehta & Rastegari (2023), and EfficientVMamba-S Pei et al. (2025). Among them, MobileViTv2 employ separable self-attention, while EfficientVMamba is a SOTA lightweight ViM. As shown in Table 4, despite similar FLOPs between VMINet and EfficientVMamba, both formulations of VMINet demonstrate lower latency. We attribute this primarily to insufficient GPU utilization in EfficientVMamba's SSM module during shorter sequence processing. In terms of performance, the recurrent formulation of VMINet-XS (VMINet-XS-R) slightly outperforms the matrix formulation (VMINet-XS-M). We believe this is due to the recurrent formulation's enhanced ability to utilize local information across different scales. However, considering the performance-efficiency trade-off, the matrix formulation of VMINet remains the preferable choice.

Table 4: Comparison of efficient models on ImageNet-1K. The latency is evaluated on an A6000 GPU with a batch size of 1.

| Method | FLOPs (G) | Latency (ms) | Top-1 (%) |
|---|---|---|---|
| Vim-Ti (ICML 2024a) | 1.5 | 2.6 | 76.1 |
| MobileViTv2-1.0 (TMLR 2023) | 1.8 | 2.3 | 78.1 |
| VMINet-XS-M **(ours)** | 1.4 | 1.8 | 78.6 |
| EfficientVMamba-S (AAAI 2025) | 1.3 | 2.4 | 78.7 |
| VMINet-XS-R **(ours)** | 1.4 | 2.1 | 78.8 |

**Impact of mask matrices.** For matrix-form VMI-SA, the mask matrix $M$ provides positional information for the context vector $\mathbf{c_v}$ while introducing an inductive bias regarding the importance of tokens. Specifically, let $X \in \mathbb{R}^{(L,C)}, W^1 \in \mathbb{R}^{(C,D)}, W^2 \in \mathbb{R}^{(C,D)}, Q = XW^1, K = XW^2, M \in \mathbb{R}^{(L,D)}$. For any element $e_n$ in $\mathbf{c_v}$:

$$e_n = \sum_{t=1}^{L} \alpha_t \cdot M_t \odot Q_t \odot K_t$$

$$= \sum_{t=n}^{L} \sum_{i=1}^{C} \sum_{j=1}^{C} \alpha_t W_{i,n}^1 W_{j,n}^2 X_{t,i} X_{t,j} \tag{11}$$

It is clear that $e_n$ encodes the $n$-th token and all subsequent tokens in the sequence, which implies that tokens with higher indices are encoded more frequently. To balance the importance of each token, the most straightforward method is to remove the mask matrix. However, this leads to a significant performance degradation, with the accuracy dropping from 78.6% to 76.5%, primarily due to the loss of multi-scale token dependencies and positional information. Similar to triangular matrices, banded matrices and block diagonal matrices are also sparse matrices. Using them as mask matrices can provide positional information for $c_v$ while partially alleviating the issue of encoding imbalance. The forms of these matrices are illustrated in Eq. (12).

$$
M^1 = \begin{bmatrix} 1 & \cdots & 1 & & & & \\ \vdots & \ddots & \vdots & \ddots & & & \\ \vdots & & 1 & & 1 & & \\ \vdots & & \vdots & \ddots & \vdots & & \\ 1 & & & & 1 & & \\ & \ddots & \vdots & & \vdots & & \\ & & 1 & \cdots & 1 & \end{bmatrix}, \quad M^2 = \begin{bmatrix} 1 & \cdots & 1 & & & & \\ \vdots & \ddots & \vdots & & & & \\ 1 & \cdots & 1 & & & & \\ & & & \ddots & & & \\ & & & & 1 & \cdots & 1 \\ & & & & \vdots & \ddots & \vdots \\ & & & & 1 & \cdots & 1 \end{bmatrix} \quad (12)
$$

Here, we only discuss two specific cases: Let $M^1, M^2 \in \mathbb{R}^{(L,D)}$, $B = \min(L, D)$, where $M^1$ has a bandwidth of $B/2$, and $M^2$ consists of $B/2$ sub-block matrices, each of size $2 \times 2$.

Table 5: Ablation on the impact of different mask matrices.

| Form of the Mask Matrix | Top-1(%) |
|---|---|
| Baseline | 76.5 |
| + Block Diagonal Matrix | 77.4 |
| + Banded Matrix | 78.6 |
| + Lower Triangular Matrix | 78.6 |
| + Hybrid Mask Matrix | 78.9 |

We use VMINet-XS without the mask matrix as the baseline model and apply different types of mask matrices to it separately. As shown in Table 5, even when using a highly sparse block diagonal matrix as the mask matrix, the model performance still shows a significant improvement. Experimentally, there is no difference in performance when using a banded matrix or a lower triangular matrix as the mask matrix. In addition, we explore the hybrid use of mask matrices. Specifically, the VMI-SA blocks in Stage 1 and Stage 2 use a banded matrix as the mask matrix, while the VMI-SA blocks in Stage 3 and Stage 4 alternately use lower triangular and banded matrices as the mask matrices. The experimental results demonstrate that the hybrid use of different types of mask matrices can achieve better performance. We speculate that carefully designed mask matrices can further enhance the performance of VMINet, and both structural design and parameterization are promising research directions.

**Effectiveness of VMI-SA.** Setting aside the design philosophy, due to structural similarities, a reasonable suspicion is that the superior performance of VMINet may primarily be attributed to the introduction of depthwise separable convolutions. As shown in Figure 4, for VMINet-S, after removing attention-related operations such as element-wise matrix multiplication and context vector generation, VMI-SA degenerates into a block similar to a ConvNeXt block Liu et al. (2022). Although this slightly reduces the number of parameters and computational complexity, the accuracy decreases from 80.5% to 78.7%.

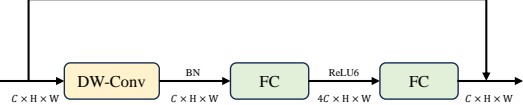

Figure 4: The VMI-SA after removing attention-related operations. It can be observed that it shares the same overall structure as the ConvNeXt block, but differs in normalization methods and activation functions.

## A.4 Additional Experimental Results

**Visualization.** We use Grad-CAM Selvaraju et al. (2020) to visualize the results of our VMINet-XS and Vim-Ti Zhu et al. (2024a) trained on ImageNet-1K. As shown in Figure 5, the activation regions of Vim in the maps are more scattered than those of VMINet, and some background areas located at the edges of the image are also activated. Although VMINet also activates some areas outside the classification objects, these regions generally contain certain semantic object information, such as the red helmet.

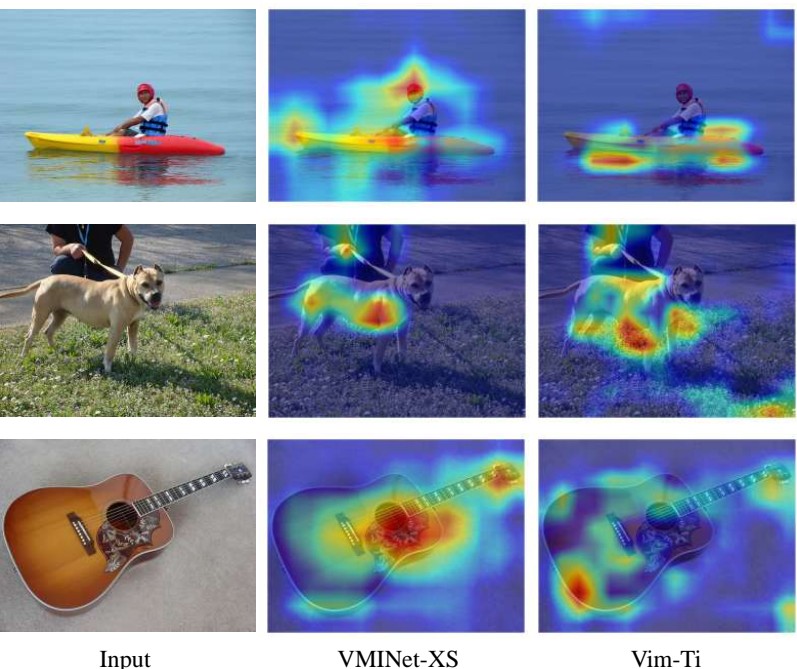

|            Input            |         VMINet-XS         |           Vim-Ti          |

Figure 5: Grad-CAM activation maps of the models trained on ImageNet-1K. The visualized images are from validation set.

Table 6: Results of semantic segmentation on ADE20K.

| Backbone | Params | mIoU |
|---|---|---|
| ResNet-50 He et al. (2016) | 67M | 40.7 |
| Vim-Ti Zhu et al. (2024a) | 34M | 41.0 |
| VMINet-XS **(ours)** | 34M | 42.7 |
| Vim-S Zhu et al. (2024a) | 57M | 44.1 |
| Swin-T Liu et al. (2021) | 60M | 44.4 |
| VMINet-S **(ours)** | 47M | 44.8 |
| ConvNeXt-T Liu et al. (2022) | 60M | 46.7 |
| VMamba Liu et al. (2024) | 60M | 47.9 |
| VMINet-B **(ours)** | 58M | 48.2 |
| VMIFormer **(ours)** | 60M | 48.8 |

**Results of Semantic Segmentation.** The results are presented in Table 6. Compared with Vim Zhu et al. (2024a), VMINet and VMIFormer once again demonstrate higher accuracy and outperforms models such as ResNet He et al. (2016), SwinLiu et al. (2021), ConvNeXt Liu et al. (2022), and VMamba Liu et al. (2024), further validating the effectiveness of VMI-SA.

