# OpenReview forum: "A Separable Self-attention Inspired by the State Space Model for Computer Vision"
_ICLR.cc/2026/Conference — Submitted to ICLR 2026_

### Official Review · Reviewer_21TC · 2025-10-21

**Soundness:** 3
**Presentation:** 3
**Contribution:** 3
**Rating:** 6
**Confidence:** 4

**Summary:**

This paper analyzes the strengths and weaknesses of separable self-attention, and based on which, proposes a new family of separable attention based model, namely VMI-SA. Comparing with previous methods, the main innovations of VMI-SA include:

1. The attention blocks in VMI-SA apply element-wise multiplication to replace the traditional matrix multiplication

2. Context vectors are introduced to replace attention matrices.

3. Depth-wise conv is used to introduce local spatial correlations before the element-wise multiplication operation.

Based on the ideas above, VMINet and VMIFormer are designed. Experimental results on image classification and object detection tasks show that the proposed method has comparable or better performance comparing with several recent proposed CNNs and Transformers.

**Strengths:**

1. The theoretical basis of the proposed method is carefully proposed.

2. The process of the designing of VMI-SA is well-presented. Each main component, such as element-wise multiplication, context vector, and MAMBA inspired attention, are analyzed detailedly.

3. The experiments cover many main-stream ViT and CNN models, thus prove the advantages of the method.

**Weaknesses:**

1. Some arguments need to be clarified. For instance, in 3.2, the authors firstly mentioned that "The higher the
rank of the attention matrix, the more attention information it contains, and the richer the feature diversity." Which implies that an attention matrix with higher rank may provide some benefits on feature extraction. After that, Eq.7 shows that the rank of context vector is less or equal with min{L, D}. Then the authors argued that the attention information in softmax(Q)⊙K is not only less abundant but also severely homogenized. Here, it seems like one benefit of context vector is to lower the rank of attention matrices. This is a conflict with the previous context. Moreover, in 3.3.2, the authors again mentioned that we need to enhance the rank of the attention matrix in the proposed method.

2. Some tiny problems that may be improved. For instance, in Figure 2, it is better to mark some important features of the model, such as Q, K, and context vector.

**Questions:**

My questions are proposed in the part "Weaknesses".

---

> ### Author Response · Authors · 2025-11-17
> **We appreciate the reviewer's positive feedback on our work**
>
> We are greatly encouraged by the reviewer's overall positive assessment of our work and will revise the manuscript according to their valuable feedback. Below, we address the reviewers' request to clarify the issue regarding the rank of attention matrices (or their counterparts).
>
> **Low rank is an inevitable consequence of the performance-efficiency trade-off, not a deliberate design choice.**  Quadratic-cost attention mechanisms (e.g., Softmax self-attention) construct complete fully-connected graphs by explicitly computing similarities between all token pairs. In contrast, linear-cost attention mechanisms (e.g., separable self-attention) must sacrifice representational capacity for efficiency, employing various techniques to compress information and avoid full pairwise computation. Consequently, a fundamental difference in expressive capacity exists between these two types of attention mechanisms, which can be measured by the rank of the attention matrix.
>
> As stated in Section 3.2, for Softmax self-attention: $\mathrm{rank}(softmax(QK^\mathrm{T}))=L$.
>
> For separable self-attention: $\mathrm{rank}(softmax(Q)\odot K) \leq \mathrm{rank}(K) \leq \mathrm{min}\\{ L,D\\}$.
>
> Since $D<<L$, we have $\mathrm{rank}(softmax(Q)\odot K) \leq \mathrm{rank}(K) \leq D$. Clearly, the rank of attention matrices based on element-wise product is bounded by $D$, while the original separable self-attention does not reach this upper bound (as $\mathrm{rank}(softmax(Q)\odot K)=D$ cannot be guaranteed in most cases).
>
> In turn, increasing the rank of the attention matrix enhances the representational capacity of the attention mechanism, thereby improving performance. As we note in Section 3.2: "FLatten Transformer … visualized changes in the rank of attention matrices. Experiments demonstrate that this enhancement enables the attention matrix to achieve full rank, resulting in significant performance improvements for the model. This indicates a positive correlation between the attention matrix rank and model performance."

---

> > ### Comment · Reviewer_21TC · 2025-11-18
> >
> > Thank you for your response. My concerns about this paper have been solved, and I would like to keep my score unchanged.

---

### Official Review · Reviewer_uDKL · 2025-11-01

**Soundness:** 4
**Presentation:** 3
**Contribution:** 3
**Rating:** 4
**Confidence:** 4

**Summary:**

This paper proposes a variant of  separable self-attention method to incorporate correlation between tokens, which a basin SSA lacks. The authors incorporates three components: SSA, depthwise convolutio, and mask matrix to enhance the rank. This paper shows somewhat strong performance.

**Strengths:**

The evaluation is quite convicing. The comparison with ViM models shows that VMiNet and VMIFormer achieve superior performance over ViM variants.

Also, the ablation study of mask in Appendix demonstrates the importance of mask operation.

**Weaknesses:**

It is not clear what the authors really adopt from SSM to this proposed model. The explation between Eq. 9 and Eq. 10 in not clear.
Also, the efficiency analysis is too limited. Efficient VMamba shows the least FLOPS with longer latency and the explatnion is "nsufficient GPU utilization in EfficientVMamba’s SSM module during shorter sequence processing." Does it mean the results would be different on longer sequences?
Also, the comparison does not include Flatten Transformer.

**Questions:**

1. Please clarify what exactly the inspiration from SSM is and the logic behind Eq. 9 and Eq. 10.
2. Please add comparison with Flatten Transformer.
3. Please include Top-5 accuracy.

* please go over equations. For example, in Eq.5 == and != should be $-$ and $\neq$.

---

> ### Author Response · Authors · 2025-11-16
> **We Thank the Reviewer for the Comments**
>
> First, we thank the reviewer for acknowledging the soundness, presentation, and contribution of our work. The reviewer's feedback is invaluable for improving the quality of our manuscript, which we will revise accordingly. Next, we will highlight some details in the manuscript that may have been overlooked and provide additional experimental results to address the reviewer's concerns. We note considerable overlap among the points raised, and thus have structured our responses collectively.
>
> On the inspiration from SSM for our VMI-SA:
>
> From design philosophy to specific functional form, our VMI-SA is inspired by SSM. The success of Vision Mamba inspired us to recognize that carefully designed causal models can also be effectively applied to 2D visual information modeling. The S4 model (a classic SSM), i.e., Equation 3 in the manuscript, compresses and maintains historical context information in a fixed-length vector (hidden state $h_i$​), which is similar both in form and function to the context vector of separable self-attention. Unlike earlier models employing separable self-attention, Vision Mamba generally integrates more seamlessly with advanced architectures and achieves superior performance across various scales. This led us to investigate whether introducing causality could enhance the performance of separable self-attention.
>
> We approached this from the perspective of matrix rank to theoretically analyze the value of introducing causality into separable self-attention. In Section 3.3.2, we note: "Next, we consider how to enhance the rank … by referring to Eq. (3)."
>
> Architecturally, while building upon SSM, we introduced optimizations resulting in the recurrent formulation of VMI-SA (Eq. 9). In Section 3.3.3, we state: "Similar to Eq. (3), we restrict the receptive field.…"
>
> On the rationale behind Eq. 9 and Eq. 10 and their connection:
>
> The logic underlying the recurrent formulation of VMI-SA (Eq. 9) is detailed in the preceding response. The matrix formulation of VMI-SA was introduced primarily to exploit the high parallelism of matrix operations, thereby further enhancing efficiency. Section 3.3.4 describes how we derive the matrix formulation (Eq. 10) from the recurrent form by altering the receptive field.
>
> Additional experiments:
>
> In response to the reviewer's concerns, we have included Flatten Transformer [1] in our comparisons and report the Top-5 accuracy for both VMINet and VMIFormer. As Top-5 accuracy was not reported in the original Flatten Transformer paper, we omit its comparison for this metric. The results demonstrate that, under comparable parameter and computation budgets, VMINet and VMIFormer outperform Flatten Transformer [1].
>
>
> **Table: Comparison of different models on ImageNet-1K.**
>
> | Method | Params (M) | FLOPs (G) | Top-1 (%) | Top-5 (%) |
> |--------|------------|-----------|-----------|-----------|
> | $\textbf{VMINet-Ti (ours)}$ | $\textbf{2}$ | $\textbf{0.3}$ | $\textbf{70.7}$ | $\textbf{90.3}$ |
> | FLatten-DeiT-T (ICCV 2023) | 6 | 1.1 | 74.1 | - |
> | FLatten-PVT-T (ICCV 2023) | 12 | 2.0 | 77.8 | - |
> | $\textbf{VMINet-XS (ours)}$ | $\textbf{7}$ | $\textbf{1.4}$ | $\textbf{78.6}$ | $\textbf{94.3}$ |
> | FLatten-PVTv2-B1 (ICCV 2023) | 13 | 2.2 | 79.5 | - |
> | $\textbf{VMINet-S (ours)}$ | $\textbf{13}$ | $\textbf{2.3}$ | $\textbf{80.5}$ | $\textbf{95.2}$ |
> | FLatten-PVT-S (ICCV 2023) | 22 | 4.0 | 81.7 | - |
> | FLatten-Swin-T (ICCV 2023) | 29 | 4.5 | 82.1 | - |
> | $\textbf{VMINet-B (ours)}$ | $\textbf{28}$ | $\textbf{4.8}$ | $\textbf{82.4}$ | $\textbf{96.1}$ |
> | FLatten-PVTv2-B2 (ICCV 2023) | 23 | 4.3 | 82.5 | - |
> | FLatten-CSwin-T (ICCV 2023) | 21 | 4.3 | 83.1 | - |
> | $\textbf{VMIFormer-T (ours)}$ | $\textbf{24}$ | $\textbf{4.2}$ | $\textbf{83.2}$ | $\textbf{96.4}$ |
>
> On model efficiency for long sequences:
>
> The inference speed of Mamba relies on the hardware-aware parallel algorithm proposed in [2], an optimization designed for the pure Mamba architecture. Since Visual Mamba models typically adopt hybrid architectures, they cannot fully leverage this optimization, thereby negating the theoretical acceleration advantage in processing images. Ablation studies of VMamba [3] (Table 9) show that the linear-complexity Vanilla-VMamba [3] underperforms the quadratic-complexity DeiT [4] across all resolutions, making comparisons between VMINet and EfficientVMamba at typical resolutions unnecessary. Mamba [2] demonstrates superior efficiency over most linear models in long-sequence tasks, suggesting that visual Mamba models may reach an efficiency inflection point at extreme resolutions—though this goes beyond the scope of our discussion.
>
> [1]Han et al. FLatten Transformer: Vision Transformer using Focused Linear Attention. ICCV2023.
>
> [2]Gu and Dao. Mamba: Linear-Time Sequence Modeling with Selective State Spaces. COLM 2024.
>
> [3]Liu et al. VMamba: Visual State Space Model. NeurIPS 2024.
>
> [4]Touvron et al.  Training data-efficient image transformers & distillation through attention. ICML 2021.

---

### Official Review · Reviewer_t7qr · 2025-11-01

**Soundness:** 2
**Presentation:** 2
**Contribution:** 2
**Rating:** 2
**Confidence:** 4

**Summary:**

This manuscript proposes a novel linear-complexity separable self-attention mechanism called Vision Mamba Inspired Separable self-Attention (VMI-SA), which draws inspiration from the SSM/Mamba while avoiding integrating any SSM blocks and featuring an attention computation process distinct from existing mechanisms. It first distill four design principles by analyzing the strengths and weaknesses of separable self-attention, then design a recurrent formulation of VMI-SA and a matrix formulation to enhance token dependency modeling and computational efficiency. Based on VMI-SA, they construct two proof-of-concept networks, VMINet and VMIFormer, and conduct fair comparisons with state-of-the-art Transformers, CNNs, and ViMs by controlling parameters, FLOPs, and encoder counts. Experimental results show that VMINet and VMIFormer achieve competitive performance in ImageNet-1K image classification, MSCOCO object detection, and ADE20K semantic segmentation, demonstrating VMI-SA’s effectiveness in balancing performance and efficiency.

**Strengths:**

1. This paper introduces an interesting model, which incorporates separate self-attention modules into the Mamba marco design.
2. The experiment are conducted on competitive benchmarks, e.g., ImageNet, COCO and ADE20K.
3. The final model have linear complexity, which is a very promising research topic to explore.

**Weaknesses:**

1. The novelty is limited. This paper incorporated the minor design in separable self-attention into the Mamba marco design, titled Mamba Inspired Separable self-Attention. It is very similar to MLLA (Mamba-Inspired Linear Attention)[1] , which incroporate the Mamba minor design into the vision transformer marco design.
2. The paper also lacks the method comparsion and performance comparsion with MLLA[1].
3. Although the authors claim this is a linear model, the performance when the token length varies is missing.
4. There is nearly no ablation in the submission. Only ablation in mask type in Tab 5. However, how each minor design affects the final result is unclear.

[1] Han et al, Demystify Mamba in Vision: A Linear Attention Perspective, in NeurIPS 2024.

**Questions:**

1. How does each detailed structure affect the final performance?

---

> ### Author Response · Authors · 2025-11-13
> **On Factual Errors in the Reviewer's Comments**
>
> We thank the reviewer for their feedback. However, we must respectfully point out significant misunderstandings regarding both our work and the cited reference MLLA [1]. Several comments also suggest that key sections of our manuscript may have been overlooked. We hope the following clarifications will enable an objective re-evaluation of our contribution.
>
> Response to Weakness 1:
>
> The reviewer's characterization is contradictory and perplexing. The reviewer state that our work "improves Mamba with separable self-attention" while MLLA [1] "improves Vision Transformers with Mamba," yet simultaneously claim the two works are "very similar." If the research object and methodology are fundamentally different, as the reviewer correctly identified, why assert similarity? This logical inconsistency undermines the critique.
> The success of Vision Mamba demonstrates that carefully designed causal models are effective for 2D visual modeling. As explicitly shown in Equations 9 and 10, VMI-SA introduces causality through specialized masking in both its recurrent and matrix forms—a core principle of our method. Conversely, MLLA [1] abandons causal modeling entirely, replacing SSM with a linear attention [2] variant (see MLLA Figure 3). This is a fundamental architectural divergence, not a minor variation.
>
> Response to Weakness 2:
>
> First, as established above, the modeling approaches are fundamentally different. Second, MLLA [1] sacrifices efficiency for performance by introducing multiple positional encodings; CPE [3] and LePE [4] introduce latent convolutional branches that increase architectural complexity and computational overhead. MLLA [1] also carefully constrains its encoder count to optimize results at specific parameter budgets.
> In contrast, VMINet and VMIFormer are designed as fair proof-of-concept models for controlled comparison, as you correctly noted in the reviewer's summary: "Based on VMI-SA, they construct two proof-of-concept networks...and conduct fair comparisons...by controlling parameters, FLOPs, and encoder counts." Direct comparison with MLLA [1] is therefore neither fair nor necessary. Nevertheless, for completeness: VMIFormer-T achieves Top-1 accuracy within 0.2% of MLLA-T under comparable parameter/FLOP budgets—a statistically insignificant difference. We are happy to include this comparison in the final version.
>
> Response to Weakness 3:
>
> The linear complexity of VMI-SA is mathematically proven in Equations 4, 9, and 10; experimental validation is unnecessary. Regarding experimental settings, our token lengths (feature map sizes) follow standard community practice and are identical to those in MLLA and other comparative works.
>
>
> Response to Weakness 4:
>
> Section A.3 contains extensive ablation studies covering: (1) mask type effects, (2) performance/efficiency trade-offs between recurrent and matrix formulation, and (3) necessity of core attention operations (element-wise multiplication and context vector generation). These experiments directly support all primary claims of the paper.
>
> Response to Question 1:
>
> This is addressed in Section A.3, as detailed in our response to Weakness 4 above.
>
>
> [1] Han et al, Demystify Mamba in Vision: A Linear Attention Perspective, in NeurIPS 2024.
>
> [2] Katharopoulos et al, Transformers are rnns: Fast autoregressive transformers with linear attention. in ICML 2020.
>
> [3] Chu et al, Conditional positional encodings for vision transformers. in ICLR 2023.
>
> [4] Dong et al, Cswin transformer: A general vision transformer backbone with cross-shaped windows. in CVPR 2022.

---

### Official Review · Reviewer_RwYd · 2025-11-03

**Soundness:** 2
**Presentation:** 1
**Contribution:** 2
**Rating:** 2
**Confidence:** 5

**Summary:**

The paper introduces Vision Mamba Inspired Separable Self-Attention (VMI-SA), a new separable self-attention mechanism drawing design principles from State Space Models, particularly Mamba. The authors propose VMINet—a prototype vision backbone built purely from stacking VMI-SA blocks and downsampling layers. Through extensive experimentation across image classification, detection, and segmentation tasks, VMINet is shown to outperform state-space-based Vim models and be competitive with strong baselines in lightweight settings.

**Strengths:**

This paper analyzes different designs principles of self-attention, vision mamba, separable self-attention, and conclude the results into four rules to guide the design of the vision models.

**Weaknesses:**

1. The involvement of causal mask does not make sense for most of the vision tasks since there are no causal hypotheses in the spatial dimension of the images and videos. That is why Mamba models [1,2,3] in vision need to define one or several complicated scanning sequence to ensure the visual signals are correctly modeled. In VMI-SA, the authors use two set of learnable gating parameters  $\alpha$s and  $\beta$s to control the proportion between the causal contexts and the direct contexts. It is of vital importance to carefully analyze how and go for different inputs and in different layers, which can provide meaningful insights on how these two types of context affect the model on vision tasks. Another important work [4] points out that the causal modeling in vision mamba models could be regarded as a forced local modeling pattern, which is also helpful. However, these analyses are missing in current submission, which fade the technical depth of the paper.

2. For the discussion part of Effectiveness of VMI-SA, the authors replace the VMI-SA block with an FC layer. However, this design choice does not resemble ConvNeXt block since the normalization layers are not the same. Current drop of the accuracy cannot support the assertion. The authors could adopt the MetaFormer [5] archictecture equipped with VMI-SA, Pooling, and Self-attention, respectively to verify the impact of the spatial modeling module.

3. The experiment results do not report the model variants in larger sizes, e.g, GFLOPs for inputs on the ImageNet-1K datasets, and models with longer sequence inputs, e.g., input resolutions. It is hard to distinguish the proposed VMINet out of the baselines such as ViM [1].

[1] Lianghui Zhu, Bencheng Liao, Qian Zhang, Xinlong Wang, Wenyu Liu, and Xinggang Wang. "Vision mamba: Efficient visual representation learning with bidirectional state space model.", ICML 2024

[2] Liu, Yue, Yunjie Tian, Yuzhong Zhao, Hongtian Yu, Lingxi Xie, Yaowei Wang, Qixiang Ye, Jianbin Jiao, and Yunfan Liu. "Vmamba: Visual state space model.", NeurIPS 2024

[3] Yang, Chenhongyi, Zehui Chen, Miguel Espinosa, Linus Ericsson, Zhenyu Wang, Jiaming Liu, and Elliot J. Crowley. "Plainmamba: Improving non-hierarchical mamba in visual recognition.", BMVC 2024

[4] Han, Dongchen, Ziyi Wang, Zhuofan Xia, Yizeng Han, Yifan Pu, Chunjiang Ge, Jun Song, Shiji Song, Bo Zheng, and Gao Huang. "Demystify mamba in vision: A linear attention perspective.", NeurIPS 2024

[5] Yu, Weihao, Chenyang Si, Pan Zhou, Mi Luo, Yichen Zhou, Jiashi Feng, Shuicheng Yan, and Xinchao Wang. "Metaformer baselines for vision.", IEEE TPAMI

**Questions:**

1. The format of the paper is a little messy with large blanks and unaligned equations.

2. The "Related Works" section is missing, making it confusing to position this paper in some lines of research and show its unique advantages.

---

> ### Author Response · Authors · 2025-11-13
> **We are disappointed by the reviewer's hasty, inconsistent, and grossly inaccurate comments**
>
> We sincerely hope that the reviewer can re-examine our work and evaluate our contributions objectively. Below, we address the weaknesses and issues raised point-by-point, drawing upon the references cited by the reviewer.
>
> Response to Weakness 1:
>
> 1. The reviewer claims that "there are no causal hypotheses in the spatial dimension of the images and videos," yet acknowledges the value of Visual Mamba [1,2,3] and notes that "Another important work [4] points out that the causal modeling in vision mamba models could be regarded as a forced local modeling pattern, which is also helpful." We find this position contradictory and perplexing.
>
> 2. The reviewer states: "That is why Mamba models [1,2,3] in vision need to define one or several complicated scanning sequence to ensure the visual signals are correctly modeled." In fact, the ablation study in VMamba [2] (Table 10) demonstrates that even the simplest unidirectional sequential scanning strategy (Unidi-Scan) achieves a Top-1 accuracy of 82.26%, compared to 82.60% for the authors' proposed Cross-Scan—a difference of merely 0.34%. This sufficiently demonstrates that Mamba's modeling of visual signals does not heavily rely on complex scanning strategies.
>
> 3. As two groups of learnable gating parameters, the values of $α$ and $β$ are highly dependent on the dataset and network architecture. Like other learnable parameters in the network, conducting an isolated quantitative analysis of them is not particularly meaningful.
>
> Response to Weakness 2:
>
> The reviewer appears to have overlooked the main focus of our "Effectiveness of VMI-SA" section, which discusses the necessity of attention-related operations (such as element-wise matrix multiplication and context vector generation). Conducting ablation experiments by simply equipping the MetaFormer [5] architecture with the VMI-SA module cannot substantiate our claims in this regard.
>
> Response to Weakness 3:
>
> The reviewer claims that "It is hard to distinguish the proposed VMINet out of the baselines such as ViM [1]." In fact, as clearly shown in Table 2 of our manuscript, VMINet-XS and VMINet-S significantly outperform the two ViM [1] variants, Vim-Ti and Vim-S. Furthermore, to ensure fair and extensive comparisons, we follow community convention by adopting 224×224 as the input resolution.
>
> Response to Question 1:
>
> This is a baffling and completely unfounded accusation. We strongly advise the reviewer to personally open the manuscript PDF file we submitted.
>
> Response to Question 2:
>
> We used "Preliminaries" instead of "Related Works" because the works we discuss are foundational (e.g., softmax self-attention) and require minimal exposition. Our focus was on presenting their formulations to help readers directly observe the similarities and differences with VMI-SA. If deemed necessary, we are happy to rename this section to "Related Works."

---

### Meta-Review · Area_Chair_TE9b · 2025-12-06

**Summary:**

This paper proposes Vision Mamba Inspired Separable Self-Attention (VMI-SA), a linear-complexity attention mechanism that bridges separable self-attention with SSM. The authors introduce both recurrent and matrix formulations of VMI-SA, and build two networks (VMINet and VMIFormer). Experiments on ImageNet classification and COCO detection/segmentation show competitive performance relative to lightweight CNNs, Transformers, and ViM variants.

Reviewers generally agreed that the paper takes an interesting direction, improving separable self-attention via masking and gating. The experiments are broad and compare against many relevant baselines. However, most reviewers found the paper lacking in clarity and completeness. Major weaknesses include unclear or incomplete technical justification for the causal masking design, contradictory core intuition in the main text, insufficient analysis of the proposed gating mechanisms $\alpha$ and $\beta$, missing comparisons with key related work such as MLLA, insufficient positioning within the linear-attention/SSM literature, and insufficient exploration of scaling properties, including larger model variants and longer sequences.

The authors provided detailed rebuttals. The clarifications could resolve part of the concerns, including the concern regarding rank from Reviewer 21TC, Top-5 accuracy and Flatten Transformer comparison from Reviewer uDKL. However, some critical concerns were not resolved. For example, the rebuttal did not strengthen the manuscript’s clarity on how VMI-SA constitutes a substantive new mechanism, which makes its novelty relative to MLLA and the broader linear-attention literature remain unclear. Reviewer RwYd’s core concern about the role and behavior of learnable gates was not addressed with quantitative evidence or analysis. Also, no evaluation on larger model sizes or longer sequences was added or clarified.

While the authors addressed some concerns during the rebuttal, the concerns regarding clarity and completeness remain. The authors are encouraged to address them in a new cycle.

**Reviewer Concerns:**

Concerns Addressed by Rebuttal:

1. Reviewer 21TC’s conceptual concerns regarding the rank argument were resolved.
2. Reviewer uDKL’s requests for Flatten Transformer comparison and Top-5 accuracy were addressed.
3. Some aspects of SSM inspiration and formulation were clarified.

Concerns Remaining After Rebuttal:

1. Reviewer RwYd: causal masking justification; lack of analysis of gates $\alpha$, $\beta$; missing large-scale experiments.
2. Reviewer t7qr: unclear novelty vs. MLLA; missing token-length experiments.
3. Reviewer uDKL: insufficient efficiency analysis and long-sequence behavior.

**Reviewer Scores:**

Reviewer RwYd: 2 → 2 (unchanged)

Reviewer t7qr: 2 → 2 (unchanged)

Reviewer uDKL: 4 → 4 (unchanged)

Reviewer 21TC: 6 →  6 (confirmed kept at 6)

---

### Decision · Program_Chairs · 2026-01-26

Reject